# ppb-Level SO_2_ Photoacoustic Sensor for SF_6_ Decomposition Analysis Utilizing a High-Power UV Laser with a Power Normalization Method

**DOI:** 10.3390/s24247911

**Published:** 2024-12-11

**Authors:** Xiu Yang, Baisong Chen, Yuyang He, Chenchen Zhu, Xing Zhou, Yize Liang, Biao Li, Xukun Yin

**Affiliations:** 1Hangzhou Institute of Technology, Xidian University, Hangzhou 311200, China; 22191214998@stu.xidian.edu.cn (X.Y.); 23191214789@stu.xidian.edu.cn (C.Z.); 23191214817@stu.xidian.edu.cn (X.Z.); 2School of Optoelectronic Engineering, Xidian University, Xi’an 710071, China; chenbaisong@xidian.edu.cn (B.C.); 22009100181@stu.xidian.edu.cn (Y.H.); liangyize@xidian.edu.cn (Y.L.); 3Chongqing Key Laboratory of Optoelectronic Information Sensing and Transmission Technology, School of Optoelectronic Engineering, Chongqing University of Posts and Telecommunications, Chongqing 400065, China; 4State Key Laboratory of Electrical Insulation and Power Equipment, Xi’an Jiaotong University, Xi’an 710071, China

**Keywords:** photoacoustic spectroscopy, gas detection, sulfur hexafluoride

## Abstract

A highly sensitive sulfur dioxide (SO_2_) photoacoustic gas sensor was developed for the sulfur hexafluoride (SF_6_) decomposition detection in electric power systems by using a novel 266 nm low-cost high-power solid-state pulse laser and a high *Q*-factor differential photoacoustic cell. The ultraviolet (UV) pulse laser is based on a passive *Q*-switching technology with a high output power of 28 mW. The photoacoustic signal was normalized to the laser power to solve the fluctuation of the photoacoustic signal due to the power instability of the UV laser. A differential photoacoustic cell can obtain a high *Q*-factor and reduce the gas flow noise in SF_6_ buffer gas. The parameters of the SO_2_ sensor system were optimized in terms of laser power and operating pressure. A 1σ detection limit (SNR = 1) of 2.34 ppb was achieved with a 1 s integration time, corresponding to a normalized noise equivalent absorption (NNEA) coefficient of 7.62 × 10^−10^ cm^−1^WHz^−1/2^.

## 1. Introduction

Sulfur hexafluoride (SF_6_) gas boasts excellent electrical insulation properties as well as remarkable arc extinguishing performance. The electrical resistance of SF_6_ is 2.5 times larger than that of nitrogen (N_2_) under the same gas pressure and temperature, the breakdown voltage is ~2.5 times, and the arcing capability is ~100 times larger than air. SF_6_ is a new-generation ultra-high voltage insulating medium material that outperforms both air and oil. SF_6_ is widely utilized in the field of microelectronics technology due to its excellent insulation and arc extinguishing performance, such as in gas circuit breakers, high-voltage transformers, gas-sealed combined capacitors, high-voltage transmission lines, and transformers. The electrical industry uses its high dielectric strength and good arc extinguishing performance as insulation materials for high-voltage switches, large-capacity transformers, and high-voltage cables. However, when this high-voltage power equipment fails, the generated arcs, sparks, etc., will cause the insulating material to decompose and chemically react with SF_6_. When the temperature of the failure point reaches 500 °C, sulfur dioxide (SO_2_), hydrogen sulfide (H_2_S), carbon tetrafluoride (CF_4_), hydrogen fluoride (HF), carbon dioxide (CO_2_), carbon monoxide (CO), and low-molecular-weight hydrocarbons are mainly produced. Therefore, the detection results of these gas decomposition products in the SF_6_ buffer gas power equipment can serve as an indicator, providing powerful support for equipment failure warnings. It has been confirmed [1] that when the concentration of sulfur dioxide in power equipment is greater than 8 ppm, the equipment must be maintained from a safety perspective [2,3,4,5,6,7].

Numerous gas sensors have been developed for monitoring SF_6_ decomposition. However, many of these sensors impose strict requirements on the detection environment and exhibit long response times. Examples include gas chromatography (GC), detection tubes, nanotechnology-based sensors, memory sensors, and sensors utilizing gas-sensitive materials [4,5,6,7,8]. In 2020, Lee et al. [9] employed a preconcentrator–gas chromatograph with a microelectron capture detector (GC-μECD) to measure SF_6_ at ambient levels, requiring a preconcentration time of 5 min. In 2021, Chu et al. [10] designed a GS microchip integrated with three gas-sensitive materials to identify multiple SF_6_ decomposition products, but the response time extended to several hundred seconds. Such prolonged response times render these sensors unsuitable for real-time monitoring applications.

Laser absorption spectroscopy [11,12,13,14,15] has been widely adopted for detecting trace gases because of its high detection sensitivity, strong selectivity, and fast response time [16,17]. In 2015, Luo et al. [1] employed a near-infrared laser and a non-resonant photoacoustic cell to monitor SO_2_, CF_4_, and CO in SF_6_ decomposition, and their minimum detection limits reached the ppm level. In 2017 and 2019, Yin et al. [18,19] used photoacoustic spectroscopy technology to detect H_2_S and CO in SF_6_ decomposition products with minimum limit sensitivities of 109 ppb and 110 ppb, respectively. In 2019, Wan et al. [20] measured CO in SF_6_ decomposition based on cavity-enhanced absorption spectroscopy with a minimum detection limit of 18 ppb. In 2021, Zifarelli et al. [21] reported an optical sensor for the detection of CO in SF_6_ gas matrix based on quartz-enhanced photoacoustic spectroscopy with a minimum detection limit of 10 ppb at 10 s of signal integration time. In 2022, Yin et al. [22] detected H_2_O in SF_6_ buffer gas with a detection limit of 0.49 ppm based on quartz-enhanced photoacoustic spectroscopy. Wei et al. [23] developed a photoacoustic sensor based on an external cavity quantum cascade laser (EC-QCL) in the same year, which can reach minimum detection limits of 70 ppb, 1.5 ppb, and 7 ppb for SO_2_F_2_, CF_4_, and SO_2_ in SF_6_ decomposition, respectively. In 2023, He et al. [24] achieved accuracies of quantitative detection for SO_2_, H_2_S, and CO in SF_6_ gas background mixture gas of 0.5 ppm, 0.1 ppm, and 0.1 ppm, respectively, by using a mid-infrared quantum cascade laser (MI-QCL). Chen et al. [25] used a high-sensitivity fiber-optic Fabry–Perot (F-P) photoacoustic gas detection system to detect H_2_S gas in the SF_6_ background with a minimum detection limit of 14 ppb in an averaging time of 200 s, by using a 1574.56 nm laser and an EDFA, in the same year. Also in 2023, Sun et al. [26] used a 1.58 μm near-infrared distributed feedback laser to achieve a ~300 ppb minimum detection limit of H_2_S in SF_6_ at an integration time of 300 ms based on light-induced thermoelastic spectroscopy. In 2024, Lv et al. [27] achieved detection limits of 0.02 ppm and 0.31 ppm for H_2_S and CO_2_ within 28 s based on CRDS combined with WDM, respectively. In addition, the concentration of SO_2_ in SF_6_ decomposition products of less than 1 ppm is expected in high-voltage power equipment for an early warning. However, in the infrared region, the absorption lines of SF_6_ and SO_2_ have a close intersection [28], which makes it a challenge to further improve the detection sensitivity in the near-infrared region. Although there is a strong SO_2_ absorption line in the mid-infrared spectral region [29], the high price of the laser light source and the interference of other gas molecules make it necessary to find other shortcuts. The existence of strong electron transition lines in the ultraviolet spectral region makes it more advantageous to detect certain gases (such as SO_2_ and H_2_S).

Photoacoustic spectroscopy (PAS) is an effective trace gas detection technique due to its advantages of high sensitivity and selectivity as well as compact detection module [30,31,32,33]. In a PAS-based sensor, the target gas absorbs the modulated excitation light and then generates an acoustic wave in a PAS cell. The acoustic wave is transformed into an electrical signal via an acoustic transducer such as a microphone or a quartz tuning fork [34,35,36]. The target gas concentration can be obtained by the amplitude of the PAS signal. In 2017, Yin et al. [28] developed a photoacoustic sensor for SO_2_ detection in SF_6_ based on a UV diode pump laser with an emission wavelength of 303 nm and an output power of 5 mW. The minimum detection limit (1σ) of 74 ppb was obtained with a 1 s integration time. In 2024, Zhao et al. [37] proposed a high-sensitivity fiber-optic photoacoustic sensor based on a UV-LED to detect SO_2_ in SF_6_ decomposition with a detection limit of 20 ppb. However, the sensor cannot run for a long time since the instability of the ultraviolet laser power has not been resolved. In addition, its detection sensitivity was limited by the lower laser power [38,39]. In 2023, Chen et al. [40] developed a compact system to stabilize laser power by adopting a photodetector, a custom-made internal closed-loop feedback controller, and a Bragg acousto-optic modulator (AOM) and used a 266 nm UV laser to detect SO_2_ in SF_6_ with a detection limit of 140 ppb. However, this method for stabilizing laser power would also greatly increase the system cost.

In this paper, a compact 266 nm UV pulsed laser based on passive *Q*-switching technology with a high output power of 28 mW was used to detect SO_2_ in SF_6_ carrier gas. The photoacoustic signal fluctuation caused by the power instability of the UV laser was solved by normalizing the signal to the laser power. A low-noise differential photoacoustic cell (PAC) with a high *Q*-factor was designed to improve the detection signal-to-noise ratio (SNR). The combination of a novel UV pulsed laser and the low-noise high-*Q* PAC offers a highly sensitive and stable SO_2_ sensor for SF_6_ decomposition detection.

## 2. Selection of Optical Excitation Source for SO_2_ Detection

According to the HITRAN database, SO_2_ molecules have six different infrared absorption bands, which are located at 2.5 μm, 3.7 μm, 4.0 μm, 7.3 μm, 8.3 μm, and 19.3 μm spectral regions, respectively. However, the interference of the SF_6_ molecular absorption line between 3.3 μm and 10 μm makes it difficult to detect SO_2_ in SF_6_ decomposition products [29]. A cross-section of the absorption line between SF_6_ and SO_2_ molecules in the 1–10 μm spectral region has been reported [28]. It turns out that the detection of SO_2_ molecules in SF_6_ decomposition products is not suitable in the infrared spectral region. The investigation found that no absorption line of SF_6_ molecules was found in the 250 nm–400 nm ultraviolet region; in order to avoid interference from the absorption line of H_2_S molecules (160 nm–250 nm), the 250 nm–320 nm ultraviolet spectrum region was selected [41]. In this spectrum region, there are indeed absorption lines of some other gases according to the HITRAN database [29], such as H_2_CO, NO_2_, O_3_, and so on. However, the sensor we designed is aimed at the monitoring of SF_6_ decomposition. SF_6_ decomposition gases mainly include SO_2_, H_2_S, CF_4_, HF, CO_2_, CO, and low-molecular-weight hydrocarbons. Therefore, there will be no other gas interferences in the 250 nm–320 nm ultraviolet spectrum region.

To reach the optimal detection band, a customized high-power semiconductor-pumped pulsed UV-band laser (Changchun New Industries Optoelectronics Technology Co., Ltd., Changchun, China, MPL-F-266 nm) with a size of 245.5 mm × 88 mm × 74 mm based on passive *Q*-switching technology was selected as the excitation light source. The geometry and physical diagram of the laser are shown in Figure 1. The diameter of the laser’s light spot is approximately 2 mm, which is much smaller than the 8 mm aperture of the photoacoustic cell. The output power of this pulsed laser can reach 28 mW at room temperature, and the emission wavelength is 266 nm. Compared with other Q-switching methods, lasers based on passive *Q*-switching technology have the advantages of low cost, a small size, and easy operation, which gives electrical fault detection instruments great advantages. Moreover, the use of a 266 nm laser can effectively avoid the interference of humidity on the detection of SO_2_ gas, because the absorption line strength of H_2_O at 37,593.98 cm⁻^1^ is nearly zero. Therefore, the influence of water molecules on the relaxation of SO_2_ gas molecules can be neglected, and there is no need to consider the interference of humidity on the sensor.

## 3. Photoacoustic Cell Design

It can be seen from Equation (1) that in the case where the optical power and the absorption coefficient of the target gas cannot be changed, only increasing the theoretical constant of the photoacoustic cell can maximize the photoacoustic signal. As expressed in Equation (2), the theoretical constant is directly proportional to the quality factor, so it becomes particularly important to use a photoacoustic cell with a high-quality factor. A differential photoacoustic cell was designed for SO_2_ detection in SF_6_ decomposition, as shown in Figure 2. The PAC with symmetrical geometry was fabricated from stainless steel. It comprised two identical parallel acoustic resonators, each 90 mm in length and 8 mm in diameter, allowing the laser beam from the UV laser to pass through the PAC. Two selected electret condenser microphones with the same sensitivity were placed on the walls in the middle of each resonator to detect the acoustic wave. Two 10 mm long buffer volumes were connected to the two resonators at both ends and constituted two identical open–open resonators. The gas-inlet and gas-outlet holes were symmetrically installed in the two buffer volumes. The PAC was sealed by two CaF_2_ windows and two rubber O-type rings. Since only one of the two acoustic resonators was irradiated by laser, the photoacoustic signal was generated only in one resonator. The signals from the two microphones were subtracted by using a custom-made differential preamplifier; thus, the flow noise, window noise, and external electromagnetic disturbances were effectively suppressed, and the signal-to-noise ratio (SNR) was improved. Figure 3 shows the values of the resonance frequency and quality factor of the low-noise differential photoacoustic cell.

For a PAS-based gas sensor, a high *Q*-factor PAC is effective in enhancing the photoacoustic signal. In general, the *Q*-factor of an acoustic resonator describes the energy losses during one period in acoustic wave propagation. For longitudinal resonance, the contribution of the losses to the *Q*-factor can be expressed as follows:$$Q=\frac{R}{d_{v}+(\gamma -1)d_{t}(1+2R/L)}$$
where γ is the ratio of the specific heats, $d_{v}$ and $d_{t}$ are the viscous and thermal boundary layer thicknesses, and R and L are the radius and length of the resonator. The theoretical value of the *Q*-factor can be calculated by the physical constants of the N_2_ and SF_6_ buffer gases at 20 °C and 1 atm. For a cylindrical resonator with a length of 90 mm and a diameter of 8 mm, the theoretical *Q* values of 38 and 81 were calculated for N_2_ and SF_6_ buffer gases. Figure 3 shows the measured frequency response curves of the PAC in the experiment when filled with 50 ppm SO_2_/N_2_ and 50 ppm SO_2_/SF_6_. The *Q*-factor can be experimentally obtained from the ratio of the resonance frequency to the full width at half maximum (FWHM) of the resonance profile. The measured resonance frequencies of the PAC were $f_{N_{2}}=1815\;Hz$ and $f_{{SF}_{6}}=699\;Hz$ in N_2_ and SF_6_, with *Q*-factors of $Q_{N_{2}}=27$ and $Q_{{SF}_{6}}=85$, respectively. The experimentally obtained *Q*-factors were in excellent agreement with the theoretical values. The *Q*-factor of the acoustic resonator in SF_6_ is much larger than that in N_2_, so the PAC can enhance the photoacoustic signal better in SF_6_.

## 4. The Establishment of the Sensing System

A photoacoustic sensor system based on an ultraviolet pulse laser and a low-noise differential photoacoustic cell was used to detect SO_2_ gas from SF_6_ decomposition products. The schematic of the sensor system is shown in Figure 4. The laser was driven by a control module from the manufacturer. However, because the repetition frequency of the pulses released by the laser based on passive Q-switching technology is uncontrollable, the repetition frequency fluctuates with the change in pump energy, which cannot meet the requirements of precise requirements for repetition frequency. Therefore, an optical chopper was employed to modulate the intensity of the pulsed laser beam so that the frequency of the beam passing through the optical chopper is always fixed at the resonance frequency of the PAC. So, a function generator (Tektronix, Inc., Biverton, OR, USA, Model AFG 3022) was employed to generate a square wave with a frequency equal to the longitudinal mode vibration frequency of the photoacoustic cell to drive the optical chopper (Thorlabs Inc., Newton, NJ, USA, model MC2000B) to modulate the laser beam intensity, and the reference signal of the function generator was transmitted to the lock-in amplifier (Stanford Research Systems, Sunnyvale, CA, USA, Model SR830). The time constant and filter slope were set to 1 s and 12 dB/cot, respectively, and the corresponding detection bandwidth reached 0.25 Hz. After the laser beam passed into a customized differential PAC, two identical electret microphones (Primo Microphones Inc., Tokyo, Japan, model EM258) were placed in two parallel cavities with a diameter of 90 mm and an inner diameter of 8 mm to collect the generated sound signals. A preamplifier was employed to transmit the signal to the lock-in amplifier which was connected to the DAQ card (National Instrument Inc., Austin, TX, UAS, model USB-6361). The laser beam was monitored by a power meter after passing the PAC, and the acquired analog signals were sent to a computer-connected DAQ card, a self-made LabVIEW program was employed to process the collected data in order to obtain the final photoacoustic signal and invert the concentration of SO_2_ gas. In addition, due to the adsorption effect of molecules, SO_2_ molecules will adhere to the wall of a PAC at normal temperature. So, an electrothermal polyimide (PI) film working at 45 °C was coated on the PAC in order to reduce this effect.

A distribution system (Environics Inc., Tolland, CT, USA, model EN 4000) was employed to produce different concentrations of SO_2_ by using 50 ppm SO_2_:SF_6_ mixture cylinders and a pure SF_6_ cylinder. The allocated gas entered the pressure controller (MKS Instruments, Andover, MA, USA, model 649B) connected to the air inlet of the PAC through a needle valve that can control the flow rate. A mass flow meter (Alicat Scientific, Inc., Tucson, AZ, USA, model M-2SLPM-D/5M) was employed to monitor the gas flow rate. After passing through the PAC, the gas enters a membrane pump (KNF Technology, Hamburg, Germany, model N813.5ANE) and is discharged out of the room.

## 5. Evaluation of Sensor Performance

### 5.1. Photoacoustic Signal Optimization

Due to the current manufacturing technology, the power of the ultraviolet laser is unstable. According to the theoretical formula of the photoacoustic signal,
(1)S=PCKα
where P represents the power of the incident laser in the PAC, C is the target gas concentration, α is the absorption coefficient of the detected gas, and K is the theoretical constant of the PAC, which is expressed as follows:(2)$$K={A_{Q}R}_{m}f_{n}$$
where $A_{Q}$ represents the quality factor coefficient of the PAC, $R_{m}$ is the sensitivity of the PAC microphone, and $f_{n}$ represents the resonant frequency of the PAC. Bringing (2) into (1), the equation can be written as follows:(3)$$S=PC{A_{Q}R}_{m}f_{n}\alpha$$

According to the formula, when the incident laser power dithers, the detected optical and acoustic signals will wobble in the same way, so that there is an error in the target gas concentration. So, a modified expression formula for the photoacoustic signal is expressed as follows:(4)$$\frac{S(t)}{P(t)}=C{A_{Q}R}_{m}f_{n}\alpha$$

Theoretical analysis indicates that the influence of the power jitter is eliminated after the normalization of the photoacoustic signal and the optical power. A power meter was used to monitor the laser output power in real time and transmit the results to a data acquisition card for calibration with a self-made LabVIEW program. As shown in Figure 5, the variation trend of the photoacoustic signal and optical power is very similar. Therefore, to mitigate the effects of instability in the high-power UV light source, we employed a normalization method, which effectively enhanced the stability and repeatability of the sensor.

### 5.2. Pressure Optimization

The pressure can change the quality factor of the PAC, the amplitude of the vibration caused by the PAC resonance, and the relaxation rate of the target gas, so the influence of the pressure on the sensing system must be optimized. The 50 ppm SO_2_:SF_6_ mixture was filled into the PAC, and the pressure range was set to 100 Torr to 700 Torr. Each increase of 100 Torr pressure value was recorded as an SO_2_ normalization signal value. A linear fitting relation (R-square ~ 0.999) is indicated and shown in Figure 6. As the pressure increases, the photoacoustic signal increases, because the greater the pressure, the stronger the relaxation between the gas molecules. In the following experiment, the pressure was selected as 700 Torr to evaluate the performance of the SO_2_ sensor.

### 5.3. Performance Evaluation

It is necessary to gradually increase the concentration of SO_2_ in the SO_2_:SF_6_ mixture to complete a detection sensitivity calculation for the performance evaluation of the photoacoustic sensor based on the ultraviolet pulse laser and the low-noise differential PAC. The different concentrations of SO_2_:SF_6_ mixture from 1 ppm to 15 ppm were proportioned and sent to the PAC. The recorded data time of each concentration reaches 200 s with an averaging time of 1 s, and the obtained response normalization signal value is shown in Figure 7a. The average value of the response signal for each SO_2_ concentration is plotted in Figure 7b to obtain the linearity of the sensor’s response to the SO_2_ concentration levels. By fitting the obtained data, an R-square result of ~0.999 was obtained.

For the signal-to-noise ratio calculation, the average power of the emitted laser was recalculated into the normalized signal because the optical power of this level has little effect on the background noise. When the PAC was filled with pure SF_6_ gas, a background noise level (1σ) equal to 0.89 μV was calculated, the response amplitude of the sensor to the mixture of 1 ppm SO_2_:SF_6_ was 0.38 mV, and the detection SNR was derived to be 427, which corresponds to a minimum detection limit of 2.34 ppb at SNR = 1. Based on the analysis of the above data, the corresponding normalized noise equivalent absorption (NNEA) coefficient is 7.62 × 10^−10^ cm^−1^WHz^−1/2^.

## 6. Conclusions

In this work, we have developed a ppb-level SO_2_ photoacoustic sensor system for the detection of SF_6_ decomposition for early warnings of electric power system failure. A compact high-power UV pulse laser emitting at 266 nm with an average output power of 28 mW was used. For the problem of laser power jitter, the analog signal of the laser output power fed back by the power meter was used to correct the photoacoustic signal. Moreover, a high *Q*-factor differential photoacoustic pool was applied to reduce the noise caused by the SF_6_ flow rate and further improve the detection limit of the sensor. A 1σ detection limit (SNR = 1) was obtained in the case of 1 s integration time reaching 2.34 ppb, and the corresponding NNEA is 7.62 × 10^−10^ cm^−1^WHz^−1/2^. The response time of the SO_2_ gas sensor is 1 s. The sensitivity of the SO_2_ sensor better meets the needs of fault warning in the power system and lays a good foundation for better field testing in the future.

## Figures and Tables

**Figure 1 sensors-24-07911-f001:**
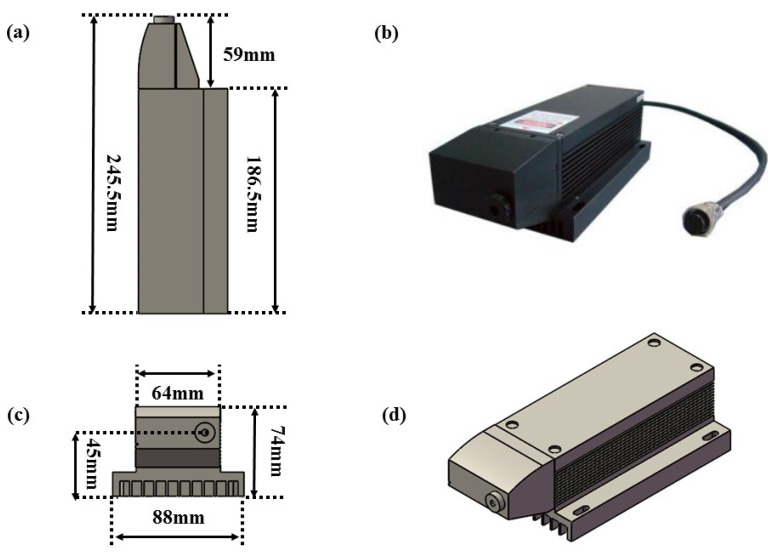
(**a**,**c**,**d**) The engineering geometry of the 266 nm UV laser. (**b**) Photograph of the 266 nm UV laser.

**Figure 2 sensors-24-07911-f002:**
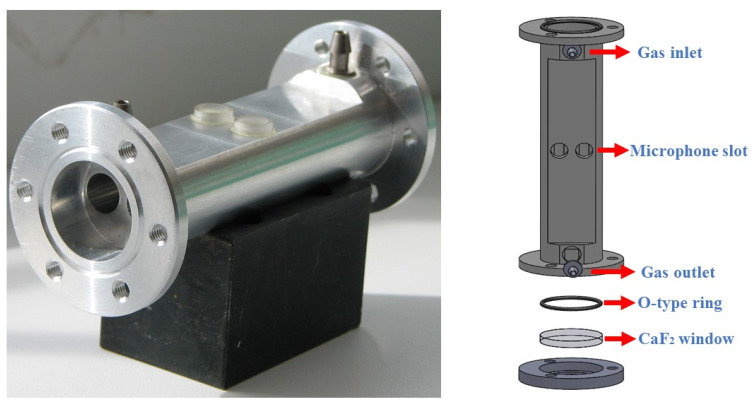
A photograph of a low-noise differential photoacoustic cell and 3D anatomical schematic diagram.

**Figure 3 sensors-24-07911-f003:**
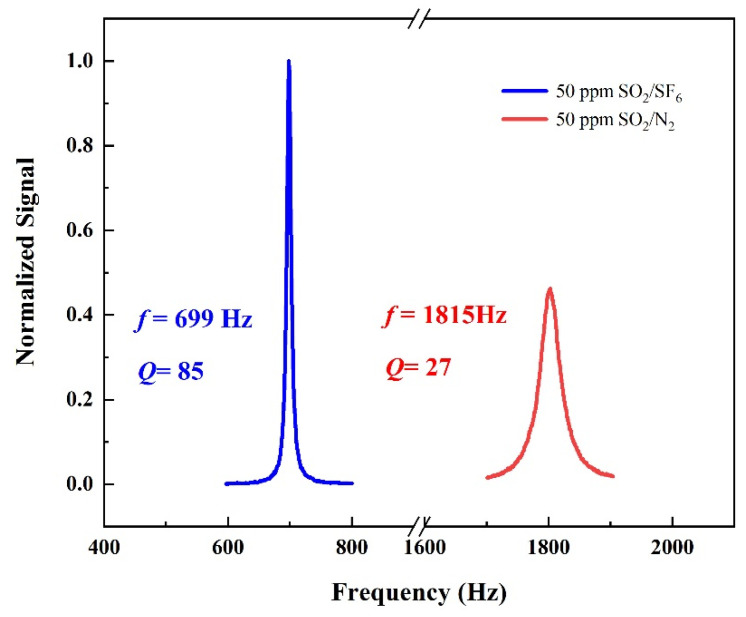
The frequency response curve of the photoacoustic cell is filled with 50 ppm SO_2_:SF_6_ mixed gas.

**Figure 4 sensors-24-07911-f004:**
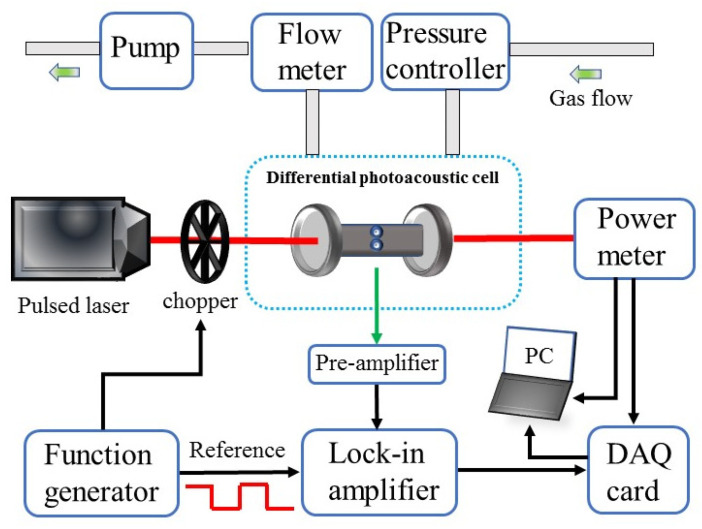
Schematic of SO_2_ photoacoustic sensor system; DAQ card: data acquisition card.

**Figure 5 sensors-24-07911-f005:**
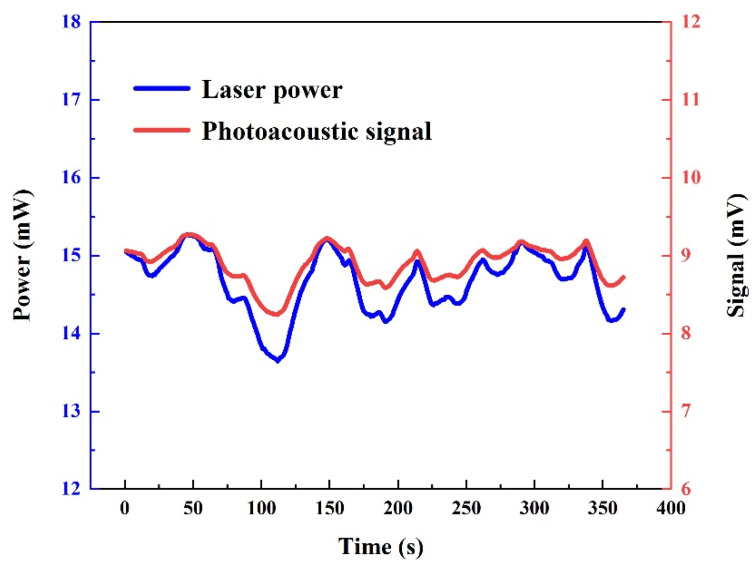
The blue curve is the power change in the laser-emitting light collected by the power meter. The red curve is the change in the photoacoustic signal of the 50 ppm SO_2_:SF_6_ mixture.

**Figure 6 sensors-24-07911-f006:**
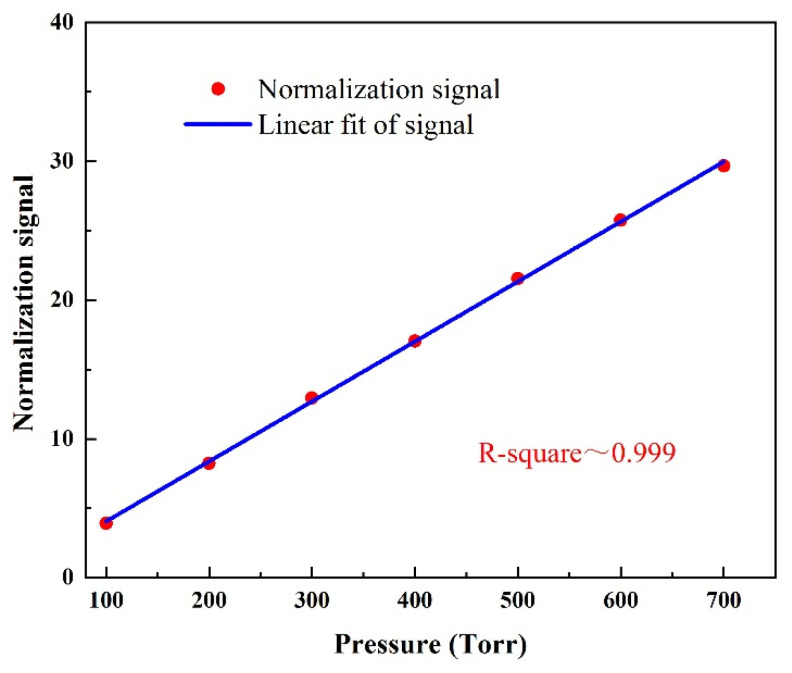
The function relation between the response amplitude and pressure of the sensor to a S0_2_:SF_6_ mixture of 50 ppm.

**Figure 7 sensors-24-07911-f007:**
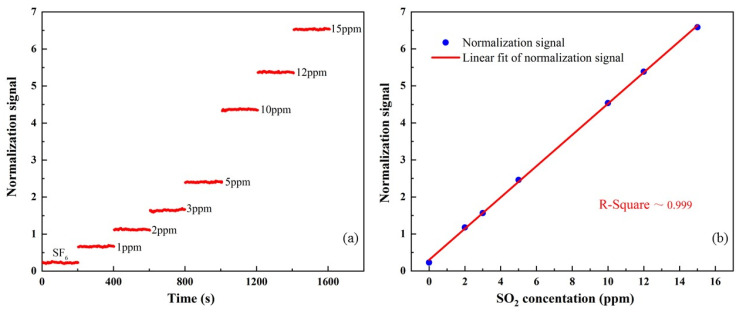
(**a**) Normalization signal amplitudes at different concentrations of SO_2_/SF_6_ mixture gas. (**b**) Normalization signal as a function of SO_2_ concentration.

## Data Availability

Data are contained within the article.

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
