# Peer review of "ppb-Level SO2 Photoacoustic Sensor for SF6 Decomposition Analysis Utilizing a High-Power UV Laser with a Power Normalization Method"

_sensors, 2024, doi:10.3390/s24247911_

Round 1

Reviewer 1 Report

Comments and Suggestions for Authors

In regard to the manuscript entitled “Ppb-level SO2 photoacoustic sensor for SF6 decomposition analysis utilizing a high-power UV laser with a power normalization method”, explaining the design and fabrication and set point of a ppm-SO2 (diluted in SF6) sensor, it is my belief that the manuscript will be subject of publication after a minor review. The whole manuscript is well explained with an adequate use of the English language. My only concern is about the reproducibility of the measurements because the authors dedicated only one paragraph to actual measurements, calculating the theoretical minimum detection limit (page 8, line 268) but did not perform measurements that support that asseveration. A plot like Fig. 7a with a set of actual measurements, where the real minimum detection limit and the resolution of the measurement system (minimum difference between consecutive measurements) are shown would greatly improve the manuscript.

Author Response

In regard to the manuscript entitled “Ppb-level SO2 photoacoustic sensor for SF6 decomposition analysis utilizing a high-power UV laser with a power normalization method”, explaining the design and fabrication and set point of a ppm-SO2 (diluted in SF6) sensor, it is my belief that the manuscript will be subject of publication after a minor review. The whole manuscript is well explained with an adequate use of the English language. My only concern is about the reproducibility of the measurements because the authors dedicated only one paragraph to actual measurements, calculating the theoretical minimum detection limit (page 8, line 268) but did not perform measurements that support that asseveration. A plot like Fig. 7a with a set of actual measurements, where the real minimum detection limit and the resolution of the measurement system (minimum difference between consecutive measurements) are shown would greatly improve the manuscript.

Response: Many thanks for the reviewer’s comments. As shown in Fig. 7(a), the 200 data points for each SO₂:SF₆ mixture concentration over 200 seconds exhibit good stability, with the normalized signal remaining nearly constant over the entire duration. We calculated the minimum detection limit based on the 1σ value of the 200 data points. Theoretical calculations indicate that the detection limit can reach 2.34 ppb. However, in practical measurements, the experimental validation of such a low detection limit is challenging due to the unavailability of standard gases at ppb levels. Nevertheless, to verify the actual measurement capability of the sensor, we conducted real-time measurements of atmospheric SO₂ using the detection system. The results are shown in Fig. 1. Since the focus of the manuscript is on measuring SO₂ in SF₆, these results were not included in the manuscript and will be published in our future work.

Fig.1 (a) Real-time SO₂ monitoring at First Laboratory, Taiyuan (Oct. 29 - Nov. 3) using the developed PAS platform (blue). (b) Atmospheric SO2 acquired by the Wucheng monitoring station of the China National Environmental Monitoring Center in the same time period.

Reviewer 2 Report

Comments and Suggestions for Authors

A sensitive ppb-level SO2 gas sensor was developed in SF6 buffer gas for failure early warning of electric power systems in this manuscript. In order to eliminate the influence of laser power jitter, the analog laser power signal was used to correct the photoacoustic signal. I believe this manuscript deserves to be published on the Sensors journal. However, some revisions are needed before publication.

1. What is the sensitivity of your used microphone? Please give the part number and manufacture.

2. The system acquires the photoacoustic signals and the optical power magnitudes through the DAQ card. How is the time synchronization of the two collected signals ensured in the experimental data analysis?

3. In Fig. 5, what concentration ratio of the SO₂:SF₆ mixture was used to measure the photoacoustic signal shown in the figure?

4. When obtaining the SO2 normalization signal value, what was the gas flow rate set inside the photoacoustic cell?

5. The sensor described in this paper is aimed at the detection of SF6 decomposition for failure early warning of electric power systems However, the actual application demonstration has never been presented in the paper. Please add relevant application demonstrations in the revised version.

Author Response

A sensitive ppb-level SO2 gas sensor was developed in SF6 buffer gas for failure early warning of electric power systems in this manuscript. In order to eliminate the influence of laser power jitter, the analog laser power signal was used to correct the photoacoustic signal. I believe this manuscript deserves to be published on the Sensors journal. However, some revisions are needed before publication.

  1. What is the sensitivity of your used microphone? Please give the part number and manufacture.

Response: Many thanks for the reviewer’s comments. The microphone used in system is Primo Microphones Inc., Japan, model EM258, whose sensitivity is -32±3 dB, which is described in part 4.

  1. The system acquires the photoacoustic signals and the optical power magnitudes through the DAQ card. How is the time synchronization of the two collected signals ensured in the experimental data analysis?

Response: We use the DAQ (Data Acquisition) card to simultaneously collect two signals, which are the photoacoustic signal and the laser power respectively, and send the two acquired analog signals to the computer. Furthermore, the self-made LabVIEW program is employed to conduct synchronous real-time processing on the two signals, thus achieving the extraction of the normalized signal.

  1. In Fig. 5, what concentration ratio of the SO₂:SF₆ mixture was used to measure the photoacoustic signal shown in the figure?

Response: The concentration ratio of SO2:SF6 mixture used to measure the photoacoustic signal shown in Fig. 5 is 50 ppm. We have modified the description of Fig. 5 as follow:

Figure 5. The blue curve is the power change of the laser-emitting light collected by the power meter. The red curve is the change of the photoacoustic signal of the 50 ppm SO2:SF6 mixture.

  1. When obtaining the SO2 normalization signal value, what was the gas flow rate set inside the photoacoustic cell?

Response: We control the flow rate by connecting a pressure controller with the air inlet of the PAC through a needle valve, and we use a mass flow meter to monitor the gas flow rate at the same time. When obtaining the SO2 normalization signal value, the gas flow rate was controlled at 50 sccm.

  1. The sensor described in this paper is aimed at the detection of SF6 decomposition for failure early warning of electric power systems However, the actual application demonstration has never been presented in the paper. Please add relevant application demonstrations in the revised version.

Response: In this manuscript our aim was to demonstrate the detection capability of the developed sensor. Test of the sensor with real SF6 decomposition sample will be the focus of a future work.

Reviewer 3 Report

Comments and Suggestions for Authors

This paper (sensors-3303844) presents a sulfur dioxide (SO2) photoacoustic gas sensor for the sulfur hexafluoride (SF6) decomposition detection utilizing a high-power UV laser. The topic is interesting for readers, but there are many issues that need to be addressed. The manuscript may be accepted after major modifications.

1.       In Introduction: It is mentioned that “Many gas sensors for SF6 decomposition monitoring have …… require a long response time”, please elaborate on the relevant detection techniques and their shortcomings, and provide corresponding examples to prove them.

2.       Are there other gas interferences in the 250-320 nm ultraviolet spectrum region?

3.       Does humidity interfere with the detection of SO2 gas, and how to eliminate the influence of humidity on gas sensor?

4.       How to calculate the signal-to-noise ratio?

5.       What is the relationship between signal-to-noise ratio and detection limit? How to calculate the detection limit of 2.34 ppb?

6.       According to experimental data, the detection limit of ppb belongs to the theoretical value. Please standardize it in Abstract, main text, and Conclusion.

7.       It is mentioned that “most of them have high requirements for the detection environment and require a long 51 response time”, what is the response time of the SO2 gas sensor in this paper?

8.       What is the repeatability of SO2 gas sensor based on ultraviolet detection technology?

9.       Most of the cited references are old.

Comments on the Quality of English Language

The English could be improved to more clearly express the research.

Author Response

This paper (sensors-3303844) presents a sulfur dioxide (SO2) photoacoustic gas sensor for the sulfur hexafluoride (SF6) decomposition detection utilizing a high-power UV laser. The topic is interesting for readers, but there are many issues that need to be addressed. The manuscript may be accepted after major modifications.

  1. In Introduction: It is mentioned that “Many gas sensors for SF6 decomposition monitoring have …… require a long response time”, please elaborate on the relevant detection techniques and their shortcomings, and provide corresponding examples to prove them.

Response: Thanks for the reviewer’s comments, we have modified the text to the manuscript as following:

Numerous gas sensors have been developed for monitoring SF₆ decomposition. However, many of these sensors impose strict requirements on the detection environment and exhibit long response times. Examples include gas chromatography (GC), detection tubes, nanotechnology-based sensors, memory sensors, and sensors utilizing gas-sensitive materials [4–8]. In 2020, Lee et al. [9] employed a preconcentrator–gas chromatograph with a microelectron capture detector (GC-μECD) to measure SF₆ at ambient levels, requiring a preconcentration time of 5 minutes. In 2021, Chu et al. [10] designed a GS microchip integrated with three gas-sensitive materials to identify multiple SF₆ decomposition products, but the response time extended to several hundred seconds. Such prolonged response times render these sensors unsuitable for real-time monitoring applications.

We added the three following references:

       [8] L. Yang, S. Wang, C. Chen, Q. Zhang, R. Sultana, Y. Han, “Monitoring and Leak Diagnostics of Sulfur Hexafluoride and Decomposition Gases from Power Equipment for the Reliability and Safety of Power Grid Operation,” Applied Sciences 14(9), 3844 (2024).

       [9] J. Lee, G. Kim, H. Lee, D. Moon, J. Lee, J. S. Lim, “Comparative study of various methods for trace SF6 measurement using GC-µECD: Demonstration of lab-pressure-based drift correction by preconcentrator,” Journal of Atmospheric and Oceanic Technology 37(5), 901-910 (2020).

       [10] J. Chu, A. Yang, Q. Wang, X. Yang, D. Wang, X. Wang, H. Yuan, M. Rong, “Multicomponent SF6 decomposition product sensing with a gas-sensing microchip,” Microsystems & Nanoengineering 7(1), 18 (2021).

  1. Are there other gas interferences in the 250-320 nm ultraviolet spectrum region?

Response: According to the HITRAN database, there are indeed absorption lines of some other gases in the 250-320 nm ultraviolet spectral region, such as H2CO, NO2, O3 and so on. However, the sensor we designed is aimed at the monitoring of SF₆ decomposition. The SF₆ decomposition gases mainly include SO₂, H₂S, CF4, HF, CO2, CO and low-molecular hydrocarbons. As described in Part 2, SO₂ will not be interfered by other SF₆ decomposition gases and SF₆ itself in the 250-320 nm ultraviolet spectral region.

  1. Does humidity interfere with the detection of SO2 gas, and how to eliminate the influence of humidity on gas sensor?

Response: The laser emission wavelength used in the sensor system is 266 nm (37593.98 cm⁻¹). According to the HITRAN database, the absorption line strength of H₂O at 37593.98 cm⁻¹ is nearly zero. Furthermore, the humidity in the GIS is very below. Therefore, the influence of water molecules on the relaxation of SO₂ gas molecules can be neglected, and there is no need to consider the interference of humidity on the sensor.

  1. How to calculate the signal-to-noise ratio?

Response: As described in part 5.3, we calculated the signal-to-noise ratio (SNR) under the concentration of 1 ppm SO₂:SF₆ mixture. The response amplitude of the sensor to the mixture of 1 ppm SO₂:SF₆ was 0.38 mV, and the 1σ noise level of 0.89 μV was calculated under the concentration of 1 ppm SO₂ in SF₆. Therefore, dividing these two parameters can result in the corresponding signal-to-noise ratio (SNR) of 427.

  1. What is the relationship between signal-to-noise ratio and detection limit? How to calculate the detection limit of 2.34 ppb?

Response: The minimum detection limit was calculated based on the average of 200 data points and signal-to-noise ratio under the concentration of 1 ppm SO₂ in SF₆. The 1σ noise level of 0.89 μV was calculated under the concentration of 1 ppm SO₂ in SF₆ and it refers to the average of 200 data points. Our experiment shows that the response amplitude of the sensor to the mixture of 1 ppm SO₂:SF₆ was 0.38 mV, based on which the corresponding signal-to-noise ratio (SNR) can be derived to be 427. By dividing this concentration by the SNR, the minimum detection limit of 2.34 ppb can be obtained.

  1. According to experimental data, the detection limit of ppb belongs to the theoretical value. Please standardize it in Abstract, main text, and Conclusion.

Response: Thanks for the reviewer’s comments, we have modified the corresponding content, stating that the minimum detection limit refers to the noise equivalent detection limit (1σ limit, SNR=1) in Abstract, main text, and Conclusion.

  1. It is mentioned that “most of them have high requirements for the detection environment and require a long 51 response time”, what is the response time of the SO2 gas sensor in this paper?

Response: The response time of the SO2 gas sensor in this paper is 1 s.

  1. What is the repeatability of SO2 gas sensor based on ultraviolet detection technology?

Response: In previous studies, photoacoustic spectroscopy gas sensors have demonstrated good repeatability. In this manuscript, the repeatability of the SO₂ sensor primarily depends on the stability of the UV light source. To mitigate the effects of instability in the high-power UV light source, we employed a normalization method, which effectively enhanced the stability and repeatability of the sensor.

  1. Most of the cited references are old.

Response: Thanks for the reviewer’s comments, we have added some new references in the manuscript:

Chen et al. [25] used a high-sensitivity fiber-optic Fabry-Perot (F-P) photoacoustic gas detection system to detect H2S gas in the SF6 background with the minimum detection limit of 14 ppb in the 200 s averaging time, by using a 1574.56 nm laser and an EDFA, in the same year. Also in 2023, Sun et al. [26] used a 1.58 um near-infrared distributed feedback laser to achieve a ~300 ppb minimum detection limit of H2S in SF6 at an integration time of 300 ms based on light-induced thermoelastic spectroscopy. In 2024, Lv et al. [27] achieved the detection limits of 0.02 ppm and 0.31 ppm for H2S and CO2 within 28 seconds based on CRDS combined with WDM, respectively.

Three references are as follows:

[25] K. Chen, N. Wang, M. Guo, X. Zhao, H. Qi, C. Li, G. Zhang, L. Xu, “Detection of SF6 gas decomposition component H2S based on fiber-optic photoacoustic sensing,” Sensors and Actuators B: Chemical 378, 133174 (2023).

[26] B. Sun, P. Patimisco, A. Sampaolo, A. Zifarelli, V. Spagnolo, H. Wu, L. Dong, “Light-induced thermoelastic sensor for ppb-level H2S detection in a SF6 gas matrices exploiting a mini-multi-pass cell and quartz tuning fork photodetector,” Photoacoustics 33, 100553 (2023).

[27] H. Lv, X. Zhang, A. Jiang, W. Qian, C. Zhang, X. Zhang, “Detection of SF6 Decomposition Components H2S and CO2 Based on WDM and CRDS,” IEEE Transactions on Dielectrics and Electrical Insulation, (2024).

Reviewer 4 Report

Comments and Suggestions for Authors

This manuscript presents a method for detecting sulfur hexafluoride (SF₆) decomposition with high sensitivity in electric power systems. The approach utilizes a novel 266 nm low-cost, high-power solid-state pulsed laser in conjunction with a high Q-factor differential photoacoustic cell. The research holds significant value for applications in the power industry. However, the following comments should be addressed before the manuscript can be accepted:

  1. In Part 2, the physical dimensions and performance specifications of the ultraviolet laser are provided. Please include the diameter of the laser's output light spot and discuss whether this diameter might cause the laser beam to interact with the walls of the photoacoustic cell, potentially affecting system performance.
  2. In Part 3, a differential photoacoustic cell is described, with its primary advantage being noise suppression and improved signal-to-noise ratio. Please elaborate on the working principle of this cell design and how it achieves enhanced noise suppression and signal-to-noise ratio improvement.
  3. Part 4 outlines the system’s composition. Please explain how the gas flow rate inside the photoacoustic cell is controlled during operation and specify the controlled gas flow rate used in the experiments.
  4. Figure 5 shows a similar trend between the photoacoustic signal and laser power; however, their ratio appears non-constant. Could this variability affect the normalization process for the photoacoustic signal and laser power? Please address this potential issue.
  5. Provide details regarding the gas concentration used in the experiment presented in Figure 5.
  6. In Part 5, please specify the interval time between consecutive measurements during the experiments.
Comments on the Quality of English Language

The quality of English in the manuscript can be further improved, making it more polished and professional.

Author Response

This manuscript presents a method for detecting sulfur hexafluoride (SF₆) decomposition with high sensitivity in electric power systems. The approach utilizes a novel 266 nm low-cost, high-power solid-state pulsed laser in conjunction with a high Q-factor differential photoacoustic cell. The research holds significant value for applications in the power industry. However, the following comments should be addressed before the manuscript can be accepted:

  1. In Part 2, the physical dimensions and performance specifications of the ultraviolet laser are provided. Please include the diameter of the laser's output light spot and discuss whether this diameter might cause the laser beam to interact with the walls of the photoacoustic cell, potentially affecting system performance.

Response: Many thanks for the reviewer’s comments. The diameter of the light spot is approximately 2 mm, which is much smaller than the 8 mm aperture of the photoacoustic cell. When the laser beam passes through the photoacoustic cell, it will not touch the cell walls, and thus will not affect the system performance. We have added the text to the manuscript as following:

The diameter of the laser’s light spot is approximately 2 mm, which is much smaller than the 8 mm aperture of the photoacoustic cell.

  1. In Part 3, a differential photoacoustic cell is described, with its primary advantage being noise suppression and improved signal-to-noise ratio. Please elaborate on the working principle of this cell design and how it achieves enhanced noise suppression and signal-to-noise ratio improvement.

Response: As described in part 3 that “the signals from the two microphones were subtracted by using a custom-made differential pre-amplifier”, the two identical acoustic resonant cavities of the differential photoacoustic cell will have the same flow noise, window noise and external electromagnetic disturbances. By using the custom-made differential pre-amplifier to subtract the signals of the two microphones, the background noise can be effectively eliminated, thereby improving the signal-to-noise ratio.

  1. Part 4 outlines the system’s composition. Please explain how the gas flow rate inside the photoacoustic cell is controlled during operation and specify the controlled gas flow rate used in the experiments.

Response: As described in part 4, we control the flow rate by connecting a pressure controller with the air inlet of the PAC through a needle valve, and we use a mass flow meter to monitor the gas flow rate at the same time. When obtaining the SO2 normalization signal value, the gas flow rate was controlled at 50 sccm.

  1. Figure 5 shows a similar trend between the photoacoustic signal and laser power; however, their ratio appears non-constant. Could this variability affect the normalization process for the photoacoustic signal and laser power? Please address this potential issue.

Response: It can be seen from Fig. 5 that the photoacoustic signal has a very strong correlation with the laser power, and their changing trends are generally consistent. Moreover, as can be seen from the results in Fig. 7, after our optimization process, that is, normalizing the photoacoustic signal to the laser power, the R-Square result after data fitting can reach 0.999. The results show that this processing method has effectively solved the photoacoustic signal fluctuation caused by the power instability of the UV laser.

  1. Provide details regarding the gas concentration used in the experiment presented in Figure 5.

Response: The concentration ratio of SO2:SF6 mixture used to measure the photoacoustic signal shown in Fig. 5 is 50 ppm. We have modified the description of Fig. 5 as follow:

Figure 5. The blue curve is the power change of the laser-emitting light collected by the power meter. The red curve is the change of the photoacoustic signal of the 50 ppm SO2:SF6 mixture.

  1. In Part 5, please specify the interval time between consecutive measurements during the experiments.

Response: In the experiment shown in Fig. 7, the measurement of the normalized signal at a certain fixed concentration is a continuous measurement that lasts for 200 s. The transition for switching between different concentrations takes about 30 s, which is to ensure that the gas concentration in the photoacoustic cell stably reaches our required concentration.

Round 2

Reviewer 3 Report

Comments and Suggestions for Authors

Please supplement the answers to questions 2, 3, 7, and 8 in the corresponding positions in this paper for readers to understand.

Author Response

Responses to Reviewer’s comments

We appreciate your comments and contributions to improving this manuscript. We have resolved all issues and revised the manuscript accordingly. Please note: reviewers’ comments are in black; our comments are in red and revised text in blue.

Please supplement the answers to questions 2, 3, 7, and 8 in the corresponding positions in this paper for readers to understand.

Response: Thanks for the reviewer’s comments, we have modified the text to the manuscript as following:

question 2:

In this spectrum region, there are indeed absorption lines of some other gases according to the HITRAN database [29], such as H2CO, NO2, O3 and so on. However, the sensor we designed is aimed at the monitoring of SF₆ decomposition. The SF₆ decomposition gases mainly include SO₂, H₂S, CF4, HF, CO2, CO and low-molecular hydrocarbons. Therefore, there will be no other gas interferences in the 250 nm-320 nm ultraviolet spectrum region.

       question 3:

       Moreover, the use of a 266 nm laser can effectively avoid the interference of humidity on the detection of SO2 gas, because the absorption line strength of H2O at 37593.98 cm⁻¹ is nearly zero. Therefore, the influence of water molecules on the relaxation of SO2 gas molecules can be neglected, and there is no need to consider the interference of humidity on the sensor.

question 7:

The response time of the SO2 gas sensor is 1 s.

question 8:

Therefore, to mitigate the effects of instability in the high-power UV light source, we employed a normalization method, which effectively enhanced the stability and repeatability of the sensor.

Reviewer 4 Report

Comments and Suggestions for Authors

I recommend accepting this paper as it presents a well-structured, thoroughly researched, and original contribution to the field. The authors have addressed a relevant and timely topic, providing both theoretical insights and practical implications.

Author Response

Many thanks to the reviewers' comments, these suggestions improve this manuscript.